# Predicting Body Weight from Birth to Old Age in Giant Pandas Using Machine Learning

**DOI:** 10.3390/ani14243694

**Published:** 2024-12-20

**Authors:** Xingyong Zhu, Jiaheng Li, Jie Gao, Jingchao Lan, Mingxi Li, Jue Deng, Wenpei Peng, Yongyou Feng, Bi Li, Huizhong Pang, Jiawen Liu, Jie Kou, Ye Wang

**Affiliations:** 1Sichuan Key Laboratory of Conservation Biology on Endangered Wildlife, Chengdu Research Base of Giant Panda Breeding, Chengdu 610081, China; zhuxingyong@foxmail.com (X.Z.);; 2Institute of Fundamental and Frontier Sciences, University of Electronic Science and Technology of China, Chengdu 611731, China

**Keywords:** giant panda, body weight, lifespan, machine learning, prediction model

## Abstract

Giant pandas are famous for having incredibly tiny babies—a newborn cub weighs only about 0.1% of its mother’s weight. Our team studied over 26,000 weight measurements from 206 zoo pandas over 22 years, including both males and females up to 37 years old. We created a day-by-day weight model for the whole life of a panda. The model helps zookeepers and veterinarians better understand if a panda’s weight is healthy for its age and can improve how we care for pandas in zoos. This is the first time anyone has mapped out panda weight patterns across their whole lifetime, giving us valuable insights into how these beloved animals grow and develop.

## 1. Introduction

The giant panda (*Ailuropoda melanoleuca*) is a flagship species of biodiversity conservation. Besides their cuteness, the unique physiological characteristics of the giant panda have also attracted the interest of scientists. While giant pandas have evolved to consume a predominantly herbivorous diet, their taxonomic classification within the Ursidae family necessitates caution, as adults and even sub-adults can exhibit aggression [1]. This behavioral characteristic poses a significant challenge to monitoring their health effectively.

Body weight, one of the easiest measurable indicators of growth and fitness in captive animals, reveals important information about their physiological state. As a species with premature birth, giant pandas have one of the largest size differences between their cub and adult stages, with newborn cubs being only 0.1% the size of their mothers [2]. While some studies have reported rapid growth in giant panda cubs, with an average weight gain of 71.3 g per day during their first 6 months [3]. The rapid and uneven weight gain characteristic of giant panda cubs presents a challenge for establishing precise weight-for-age standards across their developmental stages. For other kinds of bears, it has also been reported that the weight of adults changes with the seasons [4,5,6]. Therefore, a comprehensive understanding of lifelong growth patterns in giant pandas is essential for developing effective conservation strategies for this vulnerable species.

Despite research efforts to understand the growth patterns of giant pandas, current health management practices still rely heavily on anecdotal experience. While previous studies have examined weight changes in giant pandas during infancy [3,7,8,9], sub-adulthood [10], and adulthood [11], these studies often suffered from limited sample sizes or the weight data only being collected intermittently. The lack of the giant panda’s lifespan growth pattern largely affects health management practices, especially feeding rations and drug dosage.

Due to variations in animal care protocols and data collection methods across different institutions, breeding records of giant pandas often lack uniformity. Machine learning approaches can be particularly effective in analyzing datasets with varying standards [12]. Currently, models trained on various parameters such as images, body size, and age have been developed to predict the healthy weight of commercial species, including horses, pigs, and dogs [13,14,15]. Machine learning or deep learning algorithms facilitate the analysis of vast and complex datasets, enabling conservationists to monitor wildlife populations more efficiently and accurately [16,17]. However, the establishment of predictive models for body weight across the entire lifetime is challenging due to difficulties in data acquisition, particularly the limited number of artificially bred individuals. The giant panda has a history of artificial breeding spanning over 40 years [18], during which a substantial amount of growth data have been generated. Taking this advantage, we aim to establish a precise model for predicting the body weight of giant pandas throughout their entire lifetime. To the current knowledge, our study is the first to investigate the prediction of giant panda weight using a combination of machine learning techniques. The models used sex and age (in days) as predictors and body weight as the response variable. The results revealed that both male and female pandas exhibited seasonal weight fluctuations with significant decreases in winter, while females showed additional weight variations across reproductive periods, peaking during late breeding. After adjusting for feeding status, we established baseline weights for adults (median: female 103 kg, male 115 kg). We also found significant sex-based differences in weight velocity during ages 1096–1460 days. Our findings have important implications for the management of giant pandas in captivity, providing a valuable tool for monitoring and optimizing their development and aging physiology.

## 2. Materials and Methods

### 2.1. Ethics and Animal Welfare

This study utilized existing data from the breeding records of the Chengdu Research Base of Giant Panda Breeding, without the direct collection of new data or the use of animals. All applicable international, national, and/or institutional guidelines for the care and use of animals were strictly followed.

### 2.2. Sources of Breeding Records

We obtained weight data for giant pandas born between 1984 and 2020 from the Chengdu Research Base of Giant Panda Breeding, spanning a recording period of 2000–2022. The dataset comprised 206 giant pandas (98 males, 108 females) with 26,340 weight records, covering an age range from birth to 37 years.

### 2.3. Raw Data Processing

The accuracy of weight prediction was improved by the strategy of separate modeling on different age groups (Figure 1A). This involved dividing the dataset into three groups: 0–30 days (newborn), 31–500 days (cub to sub-adult), and over 501 days (sub-adult/adult), reflecting the different physiological stages of giant pandas.

Recognizing the potential influence of food intake on body weight measurements, particularly in sub-adult and adult pandas with substantial daily food consumption, we implemented a data adjustment procedure. It was critical for ensuring the accurate representation of real body weight, minimizing the variability introduced by weighing pandas at different points in their digestive cycle. To do this, we first fitted a curve for the weight and age for sub-adult pandas of different sexes. Then, we calculated the mean day of record days (days with weight recorded) in each month. The differences in weight between the mean day and each record day on the fitted curve were used to adjust the weight of read data for that record day. The above process was performed on the male and female sub-adult pandas separately, and an example of the above calculation process is shown in the Appendix A (“Supplementary Methods: An example for adjusting the weights of sub-adult pandas by month”). After that, the adjusted weights of the sub-adults and the actual weights in adulthood at each month of age were grouped by k-means [19] with two expected clusters (non-empty stomach and empty stomach when measuring weight). Next, the samples closest to the higher cluster center at each month of age were deleted and the left data were kept for modeling. The above pipeline is shown in Figure 1B.

### 2.4. Training and Modelling

The sex and age (in days) of the pandas were taken as independent variables and the body weight was taken as a dependent variable. The dataset was randomly split into a training set and a test set at a ratio of 3:1 to eliminate potential data leakage. Then, we evaluated five machine learning (ML) algorithms using the initial hyperparameters outlined in Appendix A. These algorithms included (1) Random Forests (RFs), (2) Extremely Randomized Trees (ETs), (3) AdaBoost (AB), (4) Gradient Tree Boosting (GBDT), and (5) XGBoost (XGB) [20,21].

We applied 5-fold cross validation to each algorithm on the training set and used the mean R^2^ values and their standard deviations as performance metrics to determine the best-performing ML algorithm. Subsequently, the hyperparameters of the optimal algorithm were tuned multiple times to optimize model performance for a first evaluation (Table 1). Next, the whole training set was retrained by the ML algorithm with the optimal hyperparameters to build a new model, which was tested on the test set for a second evaluation. The evaluation criteria were the coefficient of determination (R^2^), mean absolute error (MAE), and mean squared error (MSE). After that, a Bootstrap method was employed. This method involved retraining 1000 models on 1000 different subsets. Each subset contained a number of samples equivalent to one-fifth of the total dataset size, obtained through resampling. This process led to a third evaluation, which used out-of-bag scores to assess model performance. The models were then used to predict weights at each day of age across different sexes. The multiple evaluations were used to prove the good robustness of the models. The above pipeline is shown in Figure 1C.

### 2.5. Statistical Analysis

The Mann–Whitney U test was used to compare the weight of adult pandas of different sexes and at different periods and to compare the weight velocities of growing pandas. The analysis and the modeling were performed using Python version 3.7.0 and libraries including pandas (0.23.4), numpy (1.15.1), scikit-learn (0.19.2), seaborn (0.9.0), matplotlib (2.2.3), scipy (1.1.0), and xgboost (1.5.1).

## 3. Results

### 3.1. Lifespan Weight Dynamics in Captive Giant Pandas

Following the data filtering pipeline (Figure 1B), we fit the scatter plots of weight versus age (in days) for sub-adult pandas to equations, as depicted in Appendix A. After adjusting for weight measurements taken on an empty stomach (see Section 2 and Appendix A for details), the mean distance between paired cluster centers (representing empty and non-empty stomach states) was 16.5 ± 7.1 kg for females and 19.9 ± 7.2 kg for males. Our results were consistent with a previous study on the daily average food consumed (16.82 kg and 21.24 kg) by the individuals Feng Yi (female) and Fu Wa (male) [11]. Our analysis suggests that the observed gap between paired cluster centers likely stemmed from variations in feeding status (before vs. after feeding) during weight measurements. By removing data points closest to the higher cluster centers, we effectively minimized the influence of diet on weight recordings. This resulted in a refined dataset of 22,795 samples, as depicted in Figure 2C. The distribution of these remaining samples remained comparable to the original dataset (Figure 2A), albeit with a reduction in overall weight values. Following this adjustment, the median weight for adult females was 103 kg, while adult males exhibited a median weight of 115 kg (Figure 2D). Statistical analysis of adult panda weights (66–246 months of age) revealed significant sexual dimorphism in body size. The Mann–Whitney U test showed highly significant differences between male and female body weights in both unadjusted (*p* = 2.7 × 10^−223^) and adjusted (*p* = 2.7 × 10^−193^) groups.

Similar to other Ursidae species, including brown bears (*Ursus arctos*) [22], polar bears (*Ursus maritimus*) [23], and spectacled bears (*Tremarctos ornatus*) [24], our results suggest that giant pandas exhibit seasonal weight fluctuations. Seasonal periods were defined on the basis of natural seasons (spring (Mar~May), summer (Jun~Aug), autumn (Sep~Nov), and winter (Dec~Feb)) in Chengdu. Regardless of sex, the weight of adult giant pandas remained stable in spring, summer, and autumn but significantly decreased in winter (*p* < 0.05; Figure 3A and Figure 3C).

However, when categorized by female reproductive activity, weight change patterns differed between adult males and females. Female reproductive activities were defined as pre-breeding (Oct 1~Jan 31), early breeding (Feb 1~Mar 21), peak breeding (Mar 22~Apr 15), late breeding (Apr 16~May 31), and nonbreeding (Jun 1~Sep 30) [25]. Females exhibited peak weight during late breeding, followed by the non-breeding season. Their weight was lower in the pre-breeding and peak breeding periods, reaching the lowest point during early breeding (Figure 3B). In contrast, males showed no significant weight fluctuations (*p* > 0.05) across reproductive periods, except for a slight decrease during early breeding (*p* < 0.05; Figure 3D).

### 3.2. Modeling and Prediction

Five ML models—RF, ET, AB, GBDT, and XGB—were evaluated for their accuracy in predicting giant panda weight across different age groups. While the RF, ET, and XGB models demonstrated high performance on the training data, their lower R^2^ scores on the test set suggest potential overfitting. GBDT consistently achieved the highest mean R^2^ values across all age groups: 0.900 for 0–30 days, 0.972 for 31–500 days, and 0.823 for sub-adults/adults (Table 2). While all models demonstrated high predictive accuracy for pandas aged 31–500 days (mean R^2^ > 0.96), predicting the weight of sub-adults/adults proved more challenging, as indicated by the lower R^2^ values.

The evaluation of modeling on the training set from the dataset of sub-adults/adults showed that the performance of all the machine learning algorithms with R^2^ scores of 0.748~0.823 was significantly improved, compared to the model without the calibration of empty stomachs (R^2^ scores of 0.461~0.660, shown in Table 2), and the lower standard deviations 0.013~0.023 (compared to the model without the calibration of empty stomachs, Std of 0.021~0.040, shown in Table 2) indicated better robustness.

Hyperparameter optimization is a useful technique for improving the performance of machine learning models [26,27]. A total of 300,000 models with different combinations of hyperparameters were evaluated. The optimal hyperparameters (loss, learning rate, n estimators, and max depth) are shown in Table 3, which resulted in the highest R^2^ scores of 0.901 ± 0.010, 0.973 ± 0.001, and 0.830 ± 0.012 on the three data subsets, respectively. The above was the first evaluation, and the optimal machine learning method and hyperparameter values were determined.

The results of the second evaluation on the test sets in 0–30 days (newborn) and 31–500 days (cub to sub-adult) were, respectively, R^2^ scores of 0.883 and 0.973, MAEs of 0.081 kg and 0.895 kg, and MSEs of 0.013 kg^2^ and 1.618 kg^2^, which are shown in Appendix A. The results of the second evaluation on the test set in sub-adults/adults were an R^2^ score of 0.850, MAE of 5.2 kg, and MSE of 42.7 kg^2^, which were much better than the results of the baseline model (R^2^ score of 0.684, MAE of 8.8 kg, and MSE of 126.8 kg^2^ in Appendix A). The distribution of the predicted weight vs. actual weight is shown in Appendix A, which converged more diagonally than the distribution for the baseline model in Appendix A.

After determining the optimal values of the hyperparameters for the three age groups, the three data subsets were, respectively, used to retrain models for the prediction of daily weight. The Bootstrap method was used to supplement 1000 predicted weight data at each day of age (males with 0~11,984 days of age and females with 0~13,717 days of age). The results of the test on out-of-bag samples were R^2^ scores of 0.890 ± 0.003, 0.972 ± 0.000, and 0.810 ± 0.006; MAEs of 0.077 ± 0.001 kg, 0.904 ± 0.004 kg, and 5.7 ± 0.1 kg; and MSEs of 0.012 ± 0.000 kg^2^, 1.651 ± 0.021 kg^2^, and 53.3 ± 1.4 kg^2^ in 0~30 days of age, 31~500 days of age, and 500 days up, respectively, which formed the third evaluation. Based on the results, the bootstrapped 95% confidence intervals and standard deviations of all ages in days (0~11,984 days of age in males and 0~13,717 days of age in females) were computed to form a data table of a weight-for-age chart (Appendix A). The consistency of the R^2^ scores across all three evaluation stages showed the robustness and reliability of our GBDT models in predicting giant panda weight (Table 4).

Figure 4A and Figure 4B illustrate the predicted daily weight of giant pandas across their lifespan, separated by sex. Both male and female pandas exhibit rapid weight gain early in life, as shown in the insets highlighting the first 30 days. While both sexes reach a similar plateau in adulthood, male pandas (B) show a higher weight compared to females (A) throughout their lifespan. The shaded regions surrounding the predicted weight curves represent the 95% confidence intervals, representing the individual variability in weight among giant pandas.

To further validate the accuracy of our GBDT models, six giant pandas representing different age groups and sexes were selected as an example. Specifically, we chose one male and one female panda from each of the following age groups: 0~1 years, 1~5 years, and 5~9 years. This selection ensured a diverse sample, enhancing the robustness of our models’ validation. We compared their actual weight measurements with the weights predicted by the models. The results are presented in Figure 4C and Figure 4D. Figure 4C shows the distribution of the actual weights of the female pandas in the predicted weight distribution chart. Figure 4D does the same for the male pandas. The calculated R^2^ values for each model were as follows: 0.978 for pandas aged 0~30 days, 0.967 for those aged 31~500 days, and 0.953 for pandas aged 500 days up. These high R^2^ values indicated a strong correlation between the actual and predicted weights across all selected pandas, reaffirming the reliability of our models.

### 3.3. Weight Velocities of Growing Pandas

Understanding the weight velocities in pandas is essential for assessing their physiological and nutritional needs during critical developmental stages, as well as how sex may influence their growth trajectories. We calculated the weight velocities of growing pandas using two methods. First, we calculated the average daily weight gain from the actual weight data by taking the difference in weight between two adjacent dates and divided by the number of days between those dates. This value was assigned as the weight velocity for the midpoint of the two dates. However, due to the sparsity of the actual weight data, we also calculated weight velocities from the predicted daily weight data. In this method, the weight velocity for a given day was determined by calculating the difference between the average predicted weight for that day and the average predicted weight for the previous day.

Both methods revealed significant sex-based differences in weight velocity within the cohort of pandas aged 1096–1460 days (Figure 5). This difference in weight velocity during this crucial developmental period might be a key factor contributing to the weight discrepancy observed between male and female adult pandas. Notably, no significant sex-based differences in weight velocities were observed in other age groups of growing pandas.

## 4. Discussion

This study provides a comprehensive analysis of lifespan weight dynamics in captive giant pandas, revealing key sex-based differences and developing a robust predictive model for daily weight (Figure 4). The premature birth and rapid postnatal growth of giant panda cubs present ongoing challenges for their caretakers. Weight, as a readily obtainable physiological indicator, offers valuable insights into a cub’s health and development, especially in the management of animals with larger body sizes [28,29,30,31]. Our weight prediction model provides a powerful tool to effectively assess the physical condition of giant pandas, particularly during the critical early stages of life. By monitoring weight against the model’s predictions, it can identify potential growth concerns and make informed decisions regarding nutritional and husbandry adjustments to optimize cub development.

Our results also demonstrate that giant pandas exhibit seasonal weight fluctuations, with a significant decrease observed during winter, irrespective of sex (Figure 3). It is important to note that the captive giant pandas included in this study, unlike their wild counterparts, do not experience seasonal variations in food availability. This suggests that the observed seasonal weight changes are more likely driven by physiological adaptations and shifts in metabolic processes within the pandas themselves, rather than food limitations. This finding aligns with previous studies on other Ursidae species and warrants further investigation into the specific metabolic mechanisms underpinning these seasonal weight variations [22,23,24].

The weight velocities of sub-adult giant pandas (365–2008 days old) showed a rapid initial growth phase followed by a slower growth period (Figure 5). This pattern is consistent with previous findings by Liu et al. [10], which reported a similar rapid-then-slow weight gain trajectory in sub-adult giant pandas. Notably, male pandas exhibited significantly faster weight gain than females at around 3 years of age (Figure 5). Similar sex-specific growth patterns have been observed in humans, with males exhibiting faster growth rates than females during late childhood [32,33].

There are limitations in our study. Despite the large size of the dataset, the distribution of the weight in the pandas was not balanced across all ages. Particularly in sub-adult and elderly pandas, the insufficient sample size led to potential biases in model predictions. This imbalance in data distribution resulted in anomalous fluctuations in predictions for specific age ranges, thereby reducing the model’s reliability in some special scenarios. This uneven distribution of the weight limited the modeling and prediction of sub-adults/adults. The vacancy in the weight of males between 8k and 11k days of age (Figure 2C) reduced the accuracy of predicting weight in the elderly and the scarcity of weight data for females over 10k days of age (Figure 2D) led to abnormal fluctuations in weight. Future research should collect more data on sub-adults and adults to further improve the accuracy of model predictions. While our model demonstrates value in practice, the limited data availability means that we cannot exclude the potential for additional parameters, such as chest circumference, height, or hair characteristics, to further enhance model performance. Future studies should incorporate more comprehensive phenotypic data from elderly individuals to improve the care and management of aging giant pandas. Additionally, other multimodal data types, such as genomic and proteomic data, should be integrated to gain deeper insights into panda physiology. The combined analysis of these diverse data sources should have the potential to uncover novel patterns associated with age-related physiological changes in giant pandas.

## 5. Conclusions

This study investigated the growth pattern of giant pandas over their lifetime based on many thousands of weight data points, developed machine learning models to predict the daily weight for a reference of weight development in pandas, and analyzed sex differences in the weight gain of growing pandas. The Gradient Tree Boosting models demonstrated high predictive accuracy, with R^2^ scores of 0.883, 0.973, and 0.850 for the 0–30, 31–500, and >500-day age groups, respectively. The results of this study could be used to improve the management of giant pandas in captivity, such as by ensuring that they receive adequate nutrition and by monitoring their health.

## Figures and Tables

**Figure 1 animals-14-03694-f001:**
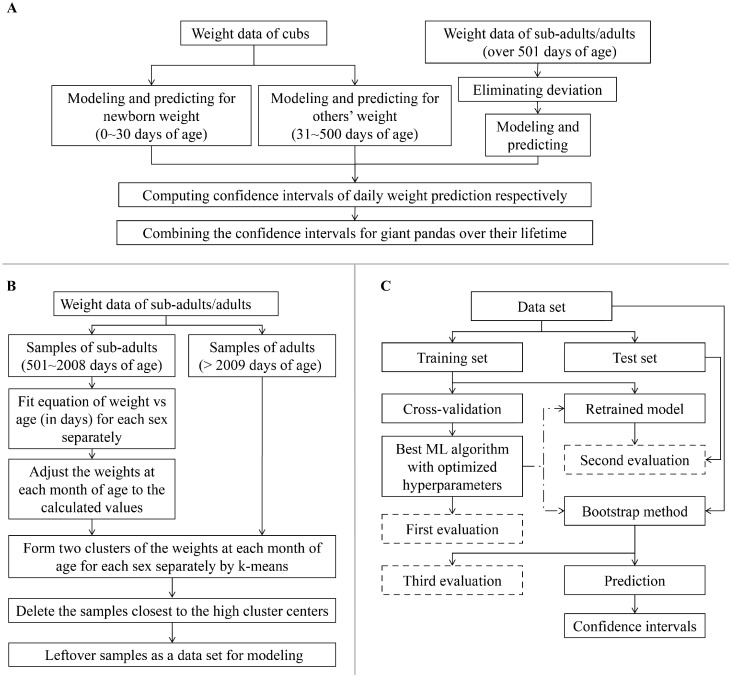
Workflow of giant panda weight prediction. (**A**) Pipeline of computing the confidence intervals of predicted daily weight of giant pandas through their entire lifetime. (**B**) Pipeline of eliminating deviation of the weight data of sub-adults/adults for calibration of weighing on an empty stomach. (**C**) Pipeline of modeling, evaluating, and predicting. ML: machine learning.

**Figure 2 animals-14-03694-f002:**
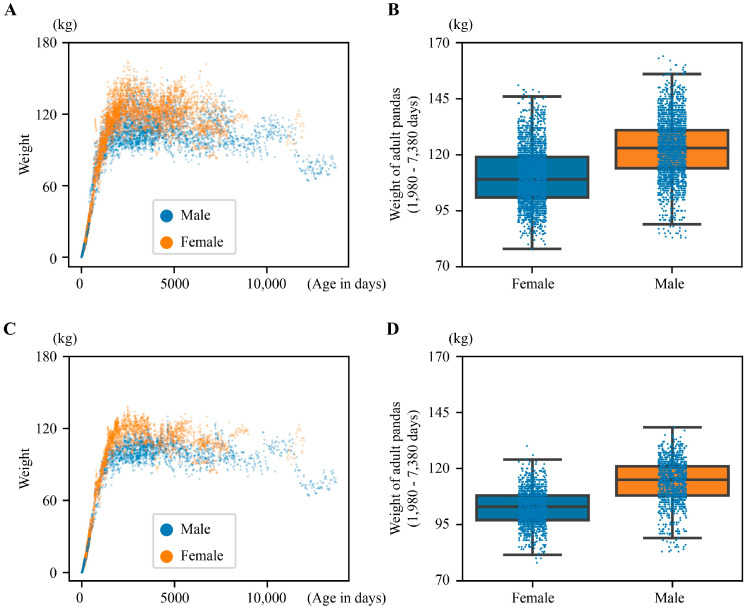
Distribution of body weight of giant pandas. (**A**) Scatter plot of original weight vs. age (in days). (**B**) Box plot of original weight of adult pandas (66~246 months of age). *p* = 2.7 × 10^−223^. (**C**) Scatter plot of weight vs. age (in days) with calibration of empty stomach. (**D**) Box plot of weight of adult pandas (66~246 months of age) with calibration of empty stomach. *p* = 2.7 × 10^−193^. Mann–Whitney U test was used for statistical testing.

**Figure 3 animals-14-03694-f003:**
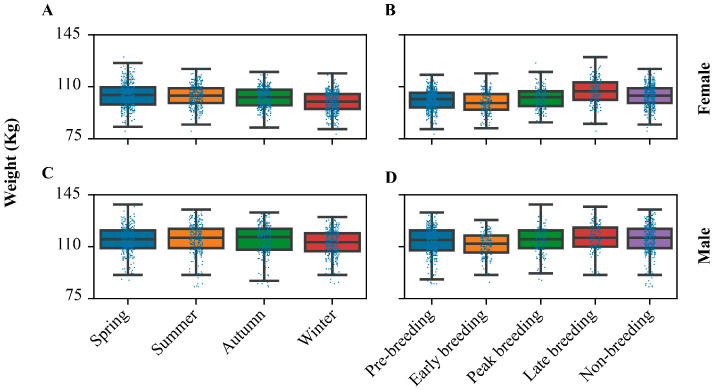
Comparison of body weight of adult pandas in different periods of a year. (**A**) Body weight of females in different periods based on natural seasons. Their weight is similar in spring, summer, and autumn (*p* > 0.05) but smaller in winter (*p* < 0.05). (**B**) Body weight of females in different periods based on female reproductive activity: early breeding < pre-breeding ≈ peak breeding < nonbreeding < late breeding (‘<’: *p* < 0.05, ‘≈’: *p* > 0.05). (**C**) Body weight of males in different periods based on natural seasons. Their weight is similar in spring, summer, and autumn (*p* > 0.05) but smaller in winter (*p* < 0.05). (**D**) Body weight of males in different periods based on female reproductive activity: early breeding < pre-breeding ≈ peak breeding ≈ nonbreeding ≈ late breeding, pre-breeding < (nonbreeding or late breeding), peak breeding < late breeding (‘<’: *p* < 0.05, ‘≈’: *p* > 0.05). Mann–Whitney U test was used for statistical testing.

**Figure 4 animals-14-03694-f004:**
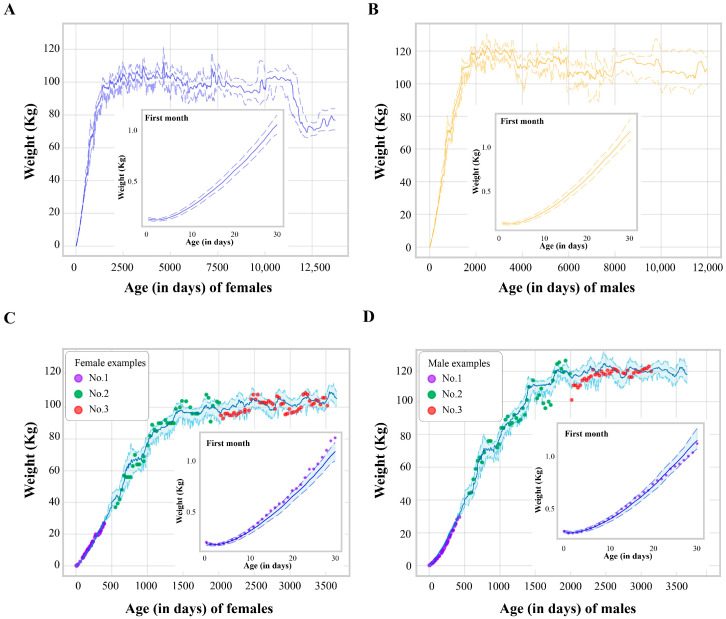
Distribution of weight vs. age after using Bootstrap method. Solid lines link median weight at each age (in days), and dashed lines define upper and lower limits of bootstrapped 95% confidence intervals. (**A**) Distribution of weight vs. age (in days) in females. (**B**) Distribution of weight vs. age (in days) in males. (**C**) Model validation using three female giant pandas from different age groups (0–1 year: No. 1; 1–5 years: No. 2; and 5–9 years: No. 3). Colored dots represent actual weight measurements, while light blue lines indicate predicted weight distributions. Inset shows detailed weight trajectories during the first month of life. (**D**) Model validation using three male giant pandas from corresponding age groups, following the same format as Figure 4C.

**Figure 5 animals-14-03694-f005:**
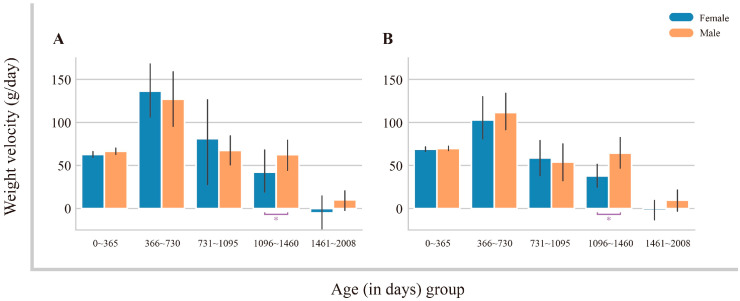
Weight velocities of growing pandas (aged 0~2008 days of age). (**A**) Statistical analysis of weight velocities calculated using actual weight data. (**B**) Statistical analysis of weight velocities calculated using predicted weight data. *p* values were computed by Mann–Whitney U test (*: *p* < 0.05).

**Table 1 animals-14-03694-t001:** Optional values of the hyperparameters of GBDT.

Hyperparameter	Option	Count
Loss	ls, lad, huber	3
Learning rate	0.05, 0.06, 0.07, …, 0.29	25
N estimators	90, 91, 92, …, 1089	1000
Max depth	2, 3, 4, 5	4

Loss: the loss function to be optimized. Learning rate: the learning rate determines the contribution of each tree to the final model. N estimators: the number of estimators, or trees, dictates the complexity of the model. Max depth: the maximum depth of the individual regression estimators.

**Table 2 animals-14-03694-t002:** Performance of ML models for weight prediction.

Age Group (Days)	Score	Model				
		RF	ET	AB	GBDT	XGB
0 to 30	Mean of R^2^	0.898	0.898	0.893	0.900	0.898
	Std	0.012	0.011	0.013	0.012	0.011
	Mean of train R^2^	0.905	0.905	0.896	0.905	0.905
31 to 500	Mean of R^2^	0.967	0.966	0.965	0.972	0.968
	Std	0.001	0.001	0.001	0.001	0.001
	Mean of train R^2^	0.977	0.977	0.966	0.975	0.977
Sub-adults/adults (With calibration of empty stomach)	Mean of R^2^	0.782	0.748	0.776	0.823	0.814
	Std	0.022	0.023	0.017	0.013	0.015
	Mean of train R^2^	0.956	0.979	0.785	0.847	0.926
Sub-adults/adults (Without calibration of empty stomach)	Mean of R^2^	0.530	0.461	0.628	0.660	0.634
	Std	0.040	0.039	0.021	0.024	0.024
	Mean of train R^2^	0.891	0.935	0.634	0.685	0.786

The evaluation results are summarized as the means and standard deviations of R^2^ scores calculated from 5-fold cross-validation and the means of train R^2^ scores on the training subsets used for model training. RF: Random Forest; ET: Extremely Randomized Tree; AB: AdaBoost; GBDT: Gradient Tree Boosting; XGB: XGBoost.

**Table 3 animals-14-03694-t003:** Optimal values of hyperparameters of GBDT and model performance.

Age Group (Days)	Loss	Learning Rate	N Estimators	Max Depth	R^2^
0 to 30	ls	0.09	95	2	0.901 ± 0.010
31 to 500	ls	0.06	91	3	0.973 ± 0.001
>500	ls	0.14	93	4	0.830 ± 0.012

**Table 4 animals-14-03694-t004:** Evaluation of GBDT model performance in weight prediction.

Age Group (Days)	Evaluation Stage	R^2^
0–30	1st	0.901
	2nd	0.883
	3rd	0.89
31–500	1st	0.973
	2nd	0.973
	3rd	0.972
>500	1st	0.83
	2nd	0.85
	3rd	0.81

First evaluation: cross-validation on the training set. Second evaluation: evaluating on the test set. Third evaluation: evaluating on the out-of-bag samples by the Bootstrap method.

## Data Availability

All original data is included in the main text and Appendix A.

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
