# Peer review of "Predicting Body Weight from Birth to Old Age in Giant Pandas Using Machine Learning"

_animals, 2024, doi:10.3390/ani14243694_

Round 1
Reviewer 1 Report
Comments and Suggestions for Authors
Congratulations to all the authors. Their modeling of the development of giant pandas is very interesting. However, since the weight distribution of pandas is not balanced across all ages, this deficiency should be addressed in future studies. Furthermore, further expansion of the discussion would make the shortcomings of the article more apparent.
Reviewer 2 Report
Comments and Suggestions for Authors
The manuscript presents a novel approach to predicting the body weight of giant pandas over their entire life span using machine learning methods. The article uses an impressive database of more than 26,000 records, making the study particularly valuable in statistical terms. Modeling with Gradient Tree Boosting yielded high accuracy predictions, which confirms the robustness of the analysis. The work is a valuable contribution to the study of panda biology and their management in breeding conditions.
The paper is the first to analyse pandas' weight growth patterns over their entire life span in such detail, providing new tools for managing their health and well-being. The results can be immediately used by keepers and veterinarians to monitor the health and adjust the diet of pandas in farms.
The use of advanced machine learning algorithms with appropriate model validation and the use of a large data set ensures the reliability of conclusions, however, the unevenness of the data (e.g. a limited number of records for older individuals) limits the generality of conclusions for these groups. It is worth highlighting this issue as a potential area for future research.
Language and spelling errors:
1. "giant panda’ lifespan" (instead "giant panda’s lifespan") [Wers 57].
2. "convince intervals" (instead "confidence intervals") [Wersy 256, 258].
3. Clarification of the expression "non-empty stomach and empty stomach" (np. "empty stomach vs. non-empty stomach") [Wers 117].
1. There are no full titles of journals, e.g. item 4 ("Blanchard, B.M.J.B.T.B.; Management. Size and growth patterns of the Yellowstone grizzly bear. 1987, 99-107").
2. Lack of consistency in the spelling of authors – sometimes full names are used, sometimes only initials (items 7, 8, 9).
3. The DOI format is not standardized. It should be written in full form, e.g. "https://doi.org/10.4238/2015.March.27.17".
4. Abbreviations of journal titles, such as "J.J.o.Z." (item 23), should be expanded in the full name of the journal.
Summarizing
The manuscript is well prepared, but before it is accepted for printing, it is recommended to make minor linguistic corrections, standardize the bibliography and better refine the illustrations. The results of the study make an important contribution to the management of giant panda populations in farms, and the method described in the article can be widely used in practice.
Recommendation: After making the suggested corrections, I recommend the manuscript for printing.
Reviewer 3 Report
Comments and Suggestions for Authors
The paper is very interesting, and looks suitable for publication. This reviewer have the next minor suggestions.
1. It is better to use percentage than fractions for a better understanding of the readers, like in line 15, authors use "1/900th".
2. ¿How was considered the diet for pandas in the used data base?
3. There is very important information in the Supplementary Materials file, about the evaluation of the algorithms of ML. This information should be in the main paper for a better explanation.
4. ¿Why it is important to measure and predict the weight of pandas?
5. Authors should add, in a table, the standard deviation of the obtained results with ML.
6. Authors should take a particular case of study of one panda, and make a graphic to compare the real measurements wit the obtained with ML, thus we can a comparison between the real data with the obtained data.
Reviewer 4 Report
Comments and Suggestions for Authors
This paper analyzed body weight records from 206 captive giant pandas collected between 2000 and 2022. The dataset comprised 26,340 measurements, including 12,314 records from 98 males and 14,026 from 108 females, with the oldest male and female being 32 and 37 years old, respectively. And this study predicted daily body weights throughout the pandas' lifespan, allowing us to establish detailed normative body weight ranges from birth to advanced age daily. This study presents the first comprehensive analysis of lifetime body weight distribution in giant pandas, enhancing our understanding of their developmental biology and informing improved body weight management strategies for captive populations. These findings have important implications for the management of giant pandas in captivity, providing a valuable tool for monitoring and optimizing their growth and development.
The main specific comments are as follows.
1. Please provide the experimental results and corresponding conclusions in the abstract as shown in Line 76-77. At the end of the introduction, explain the contribution of this paper.
2. In Line 75-79, This part of the content is recommended to be placed in the conclusion section.
3. In Line 90-93, “The dataset comprised 206 giant pandas (98 males, 108 females) with 26,340 weight records, covering an age range from born to 37 years.” Shouldn't the oldest panda be 38 years old (2022-1984)?
4. In line 97-99, ” This involved dividing the dataset into three groups: 0-30 days (newborn), 31-500 days (cub to sub-adult), and over 501 days (sub- adult/adult)”, What is the basis for dividing into three groups? Will there be a lot of data in the third group? Has the issue of data imbalance been considered? Please supplement the amount of data for each group.
5. In line 119,” the samples closest to the higher cluster center at each month of age were deleted and the left data were kept for modeling.” What are the reasons for doing these two operations? Are the right data also deteled?
6. In line 175-177, “Box plot of the weight of the adult pandas (66~246 months of age). P = 2.7×10-223. (C) Scatter plot of the actual weight vs age (in days). (D) Box plot of the weight of the actual adult pandas (66~246 months of age). P = 2.7×10-193.”, What does “P = 2.7×10-223 and P = 2.7×10-193” represent? Please introduce its meaning in the main text.
7. In section 3.13.1. Lifespan Weight Dynamics in Captive Giant Pandas, What model was used to obtain the results of this section?
8. In fig.5, Lack of predictive data for female pandas (aged 1461~2008 days of age) as shown by the red circle. And What could be the reason for the significant discrepancy between the predicted and actual results for pandas’ weight velocities (aged 366~730 days of age)?
